# Quinones as an Efficient Molecular Scaffold in the Antibacterial/Antifungal or Antitumoral Arsenal

**DOI:** 10.3390/ijms232214108

**Published:** 2022-11-15

**Authors:** Marcos Aurelio Dahlem Junior, Ronald W. Nguema Edzang, André Luis Catto, Jean-Manuel Raimundo

**Affiliations:** 1Aix-Marseille Université, CNRS, CINaM, 13009 Marseille, France; 2Department of Chemistry and Biochemistry, University of Health Science (USS), Libreville 18231, Gabon; 3Center for Exact and Technological Sciences (CETEC), University of Taquari Valley, Lajeado 95914-014, RS, Brazil

**Keywords:** quinones, biocidal properties, antibiofilm, antifungal

## Abstract

Quinone-based compounds constitute several general classes of antibiotics that have long shown unwavering efficiency against both Gram-positive and Gram-negative microbial infections. These quinone-based antibiotics are increasingly popular due to their natural origins and are used in natural beverages from herbs or plants in African, Chinese and Indian traditional medicines to treat and prevent various diseases. Quinone-based antibiotics display different bioactive profiles depending on their structures and exert specific biocidal and anti-biofilm properties, and based on recent literature, will be discussed herein.

## 1. Introduction

Nowadays, infections from bacterial and viral pathogens are a great cause of mortality and morbidity worldwide [1], and which continue to thrive despite the successful development of antiviral and antibiotic drugs because of the combination of biological mutations leading to multidrug resistances [2], efflux pumps [3], porin protein loss and decreased drug uptake [4], among others [5] and, therefore, contribute to increasing therapeutic failures. Quinone-based scaffolds are ubiquitous in nature and are found in diverse natural products such as plants, bacteria, algae and fungi [6]. For instance, they play important roles in many enzymatic and physiological systems, especially plastoquinone and ubiquinone, because of their central role as redox mediators in several electron-transfer processes in living organisms (Figure 1) [7].

Quinone derivatives have prominent applications in several areas, such as chemosensor, catalyst, dyes, energy storage, electron transfer, among others [8,9]. Furthermore, natural, or synthetic quinone derivatives are important members of drug or potential molecules in the medical and pharmaceutical industry [8,9,10]. Because of their unique pharmaceutical applications, quinones are widely used as anticancer, antioxidant, antimalarial, antimicrobial and anti-inflammatory agents [9]. Quinones have intramolecular unsaturated cyclic diketone structures and are even easily transformed into these structures. The structure of natural quinones can be mainly divided into benzoquinone, naphthoquinone, anthraquinone and phenanthraquinone. Benzoquinone is divided into 1,2-benzoquinone and 1,4-benzoquinone. 1,2-benzoquinone has an unstable instrument, causing most benzoquinones to be 1,4-benzoquinone-derived. They are commonly highlighted for their privileged structure concerning medicinal chemistry. A privileged structure can be characterized as a molecular structure or a fragment that can connect or interact with several biological receptors, allowing its application in a wide range of pharmacological activities. Such structures provide active and selective ligands for multiple targets, optimizing other functional groups [7]. In addition, this type of structure also usually presents drugs-like properties, serving as a basis for the development of new compounds [11]. The main aspects related to the synthesis of more effective drugs are in their structural characteristics and in their activity rate. In this context, quinones and its derivatives play an essential role in the development of new antibacterial drugs against various bacteria and pathogens. Quinone derivatives have shown high antibacterial activity, and also these derivatives can be used in various pharmaceuticals and clinical applications [12]. Both quinone derivatives and polymers are effective against numerous microbes in curing diseases. The presence of hydrogen as a heteroatom in its structure makes quinone more reactive in combating microorganisms. In addition, as it has an electron-rich aromatic ring, which can act in electrophilic substitution, it only enhances its ability to develop new drugs [13].

Moreover, their inherent properties make them systems of choice for industrial dyestuffs, energy storage purposes in batteries [14] and are of great of interest for the synthesis of chemotherapeutic drugs [15]. Moreover, quinone-based derivatives exhibit some toxicity whether by the formation of reactive oxygen species (ROS) or due their electrophilic natures which may prove useful as arsenal to synthetize active species for struggling microbial invasion [16]. Therefore, many natural compounds containing quinone moiety have been used to prevent or treat diseases such as cancers; for example, daunorubicin isolated from *Streptomyces coeruleorubidus* has long been used in folk medicine [17,18]. Free-radical processes (FRP) play an important role in the functioning of an organism. At the same time, its activation leads to the emergence of a number of diseases, including neurodegenerative pathologies, aging, inflammatory processes, cardiovascular problems, cancer, among others. FRPs are usually induced by reactive oxygen species, which can be generated due to a failure in the biochemical processes of oxygen utilization or as a result of external factors such as γ or UV-irradiation of the biosystems [19]. In the interaction of ROS with biomolecules under conditions of low oxygen levels, FRP can occur. In this context, Samovich et al. [20] studied the effects of quinones and azoles on the formation of steady-state radiolysis products in aqueous solutions of glycerol-1-phosphate and aqueous dispersions of 1,2-dimyristoyl-glycero-3-phosphatidyl-glycerol. The authors used 1,4-benzoquinone (**10**), 2,3,5-trimethyl-1,4-benzoquinone (**11**), tertbutyl-1,4 benzoquinone (**12**), coenzyme Q_0_ (**13**), thymoquinone (**14**) and doxorubicin hydrochloride (**15**) in their study (Figure 2). Azoles represented in the study were imidazole (**16**), metronidazole (**17**), triazole (**18**), sanazole (**19**), 1-methyl-3,5-dinitro-1,2,4-triazole (**20**) and Dinitro triazoles-sodium 3,5-dinitro-1,2,4-triazolate (**21**) (Figure 3).

The results showed that compounds with quinoid structures, including the antitumor agent doxorubicin, and azoles with nitro groups, effectively inhibit the free radical fragmentation of glycerol-1-phosphate and 1,2-dimyristoyl-glycero-3-phosphatidyl-glycerol, decreasing the radiation chemical yields of inorganic phosphate or phosphatidic acid, respectively. Thus, the research indicates the possibility of using the discovered properties of quinones, doxorubicin and nitroazoles, to provide practical solutions in oncological radiotherapy treatments and pathophysiology.

Some studies have addressed new antibacterial therapies that eradicate pathogenic bacteria using types of unconventional mechanisms. In a study by Huigens et al. [21], halogenated phenazine (HP) agents were evaluated, which induce a rapid lack of iron, and which can lead to the death of methicillin-resistant *Staphylococcus aureus* biofilms. The authors approach a microbiological study through a drug model from HP-quinone ether that acts to eliminate general iron chelation and thus releases an active HP agent through the bioreduction of a quinone. As a result, it was found that an analog of the prodrug HP-29-Q has a stable ether bond that allows the release of HP and has good antibacterial activities against some strains and other clinical isolates resistant to several drugs.

In the work developed by Hegedűs et al. [22], new aminonaphthol derivatives substituted by 2- and 1-naphthol were synthesized using naphthols, morpholine and ethyl glyoxylate as with a modified Mannich reaction. The authors tested the stabilization of bifunctional precursor compounds using aromatic ortho-quinone, testing different cyclic imines in cycloaddition (4 + 2). Plead analysis of ^1^H NMR showed the formation of a single product. The compounds were also tested on bacteria to reduce antibiotic resistance. Some of the synthesized compounds were able to inhibit the efflux pump system in methicillin-susceptible and resistant *Staphylococcus aureus* strains.

Quinone-based antibiotic compounds exert an important role in the clinical and medical arsenal to combat bacterial infections both from Gram-positive and Gram-negative strains. Interestingly, although quinone-based antibiotics present a common moiety, this feature does not confer a universal mechanism as their pharmacological action, on nucleic acids both in vitro and in vivo, has highlighted strong differences in several quinone families [23]. Furthermore, the widespread use of antibiotics as well as their overuealso led the increase of the bacterial resistance, prompting the scientific community to urgently search for new and more effective derivatives and strategies [24]. Indeed, today, microbial infections have become one of the major threats to public health, food security, and so on; therefore, is urgent to combat this by developing materials or strategies limiting or preventing bacterial proliferations and biofilm infections [25]. Awareness of biofilm colonization led to the development of an accrescent number of novel molecular structures and materials, including quinone-based compounds possessing antibiofilm features associated with biocidal properties. Despite the great efforts to discover potent biofilm inhibitors or disruptors, both their poor effectiveness for long term uses and the translation into practical applications are critical issues requiring the development of innovative therapeutic efficient tools and solutions [26].

The most predominant strategies to tackle colonization are based on the development of disrupting agents [27]. Other antibiofilm approaches based on the inhibition of bacterial adhesion or proliferation have also prompted strong interest [28]. Those include surfaces impregnated with antimicrobial and antiseptic hubs and cuffs [29], immunoprophylaxis [30], quorum sensing interference, impairment of bacteria adhesion or biofilm accumulation [31], immunotherapy, enzymatic disruption or removal biofilm [32], immunomodulation and use of nanoparticles to deliver antibiofilm agents [33]. Although some of these approaches appear promising for stamping out bacteria from infected areas, at present, it remains problematic and challenging to microbiologists, clinicians and patients who require the development of innovative therapeutic efficient tools and solutions. In addition, one important effect of the biofilm environment is to provide microbes with increased resistance to detergents and antibiotics, and in some cases, resistance can be increased as much as 1000-fold compared to their corresponding planktonic bacteria. It is therefore difficult to extrapolate planktonic bactericidal data to environmental or clinical scenarios where the majority of bacterial growth, for example on substrate surfaces, is in the form of biofilms. Thus, despite treatments, biofilms can re-grow with infection relapses and prolonged treatments may cause microbes developing an increased resistance to biocides in addition to cumulative side-effects of drugs and may have a substantial impact on the patient’s microbiota ecology. With these considerations in mind, this article aims to identify relevant properties of quinones for their use as molecular scaffolds.

## 2. Quinone-Based Antibiotics with Biocidal Properties

Quinone-based antibiotics mainly belong to the benzoquinones, naphthoquinones and anthraquinones families and exhibit significant antibacterial activities or antibiofilm properties; albeit, other quinone-based scaffolds can also be found [34]. Moreover, the presence of hydroxy groups in the ortho or adjacent positions related to the carbonyl is of crucial interest for the formation of highly stable chelate with a myriad of metal ions of biological relevance that can contribute to enlarging the scope and biocidal activities of such quinones, especially because they can influence protein structures and serve in a regulatory capacity for electron transfer, substrate recognition/binding and catalysis (Figure 4) [35].

Several natural naphthoquinones, for instance, juglone **26**, plumbagin **27**, lawsone **28** and lapachol **29** extracted respectively from *Juglans nigra, Plumbago zeylanica* roots, *Lawsonia inermis* and *Tabebuia impetiginosa* (Figure 5), have biological and pharmacological properties, and are important precursors f the synthesis of pharmaceutical products based on these organic compounds [36].

Futuro et al. [37] point to the possibility of using antifungal naphthoquinones in the fight against *Candida albicans* and its variations (*Candida glabrata, Candida parapsilosis, Candida tropicalis, and Candida krusei*). Although they indicate gaps in this progress, the authors point out the use of some naphthoquinones based on their antifungal properties (Figure 6). The mechanisms that control the biochemistry of the compounds allow for their use given the inherent similarities between fungi and human hosts, such as the fact that both are eukaryotic. 1,4-Napthoquinone Mannich adducts actively demonstrated the best antifungal activities, between 20 to 330 µg/mL.

Among the observed naphthoquinones, plumbagin also stood out for its well-established antifungal activity. 5-hydroxy-1,4-naphthoquinone, also known as juglone, can be synthesized by various methods and can be used as a starting material in synthetic transformations. It has potential use in medicine and as a biocide in organic agriculture for pest control. The structure of naphthoquinones is generally related to naphthalene (Figure 7), characterized by their two carbonyl groups at position 1,4-, being called 1,4-naphthoquinones. These carbonyl groups may be present at the 1,2-position but with considerably lower incidence [28].

Saruul et al. [38] studied the antibacterial components found in the traditional Mongolian medicinal plant *Caryopteris mongolica*. An abietane orthoquinone caryopteron A (**43**) and three its derivatives caryopteron BD (**44**–**46**) were isolated from the plant roots along with three abietanes, demethylcryptojaponol (**47**), 6α-hydroxydemethyl cryptojaponol (**48**) and 14-desoxycholeon U (**49**). Compounds **43**–**46** have C-13 methylcyclopropane substructures and compounds **44**–**46** had a hexanedioic anhydride ring C instead of ortho-quinone in **43**. The results showed that compounds **43** and **47***–***49** showed antibacterial activity against the Gram-positive bacteria *Staphylococcus aureus, Staphylococcus epidermidis, Enterococcus faecalis and Micrococcus luteus*, with structure **43** being the most promising (Figure 8).

Naphthoquinones are also capable of interacting with biological targets, establishing covalent bonds or through the ability to undergo reversible oxidation-reduction reactions. The in vivo action performed by these naphthoquinones necessarily involves bio reduction as an initial step in quinone formation. It is considered a very promising group of compounds due to the variety of biological activities it can perform, as well as the intrinsic characteristics that can be observed, which lead to chemical properties of particular importance, such as being largely oxidizing and electrophilic. [39].

In addition, aminoquinones gained considerable interest due to their antitumoral and antimalarial activities [40] and displayed effectiveness as antimicrobial drugs. For instance, several reported aminonaphthoquinones [41] have demonstrated efficient minimum inhibitory concentrations of 23.4 µg/mL and 31.3 µg/mL against *Escherichia coli* and *Staphylococcus aureus,* respectively. It has also been pointed out that the presence of a halogen atom in quinone scaffolds has a strong effect on their biological activities. Thus, in the course of a comprehensive structure-activity relationships study (SAR), the chlorinated plastoquinone-like **50** and **51** derivatives (Figure 9) [42] exhibited high minimum inhibitory concentrations of 4.88 µg/mL and 78.12 µg/mL compared to the referenced drugs Cefuroxime (9.80 µg/mL) and Amikacin (128.00 µg/mL) against the Gram-positive species *Staphylococcus epidermidis* and *Enterococcus faecalis,* respectively, whereas the unchlorinated analogues were shown to be less effective.

## 3. Quinone-Based Antibiotics with Antibiofilm Action

The use of antimicrobial agents has great prominence in the treatment of infectious diseases, as they act on the reduction of high mortality rates. However, bacteria have adapted to the presence of such inhibitory agents and become resistant to their effects, leading to a serious global public health problem.

Biofilms generated by bacteria are generally constituted by several microorganisms, ref. [43] usually embedded in a polymeric matrix. In the form of biofilm, bacteria become more resistant to antimicrobial treatments; as well, they can survive adverse conditions or even resist the immune system. This biofilm formation ability has prompted the scientific community to develop more effective drugs for treatments. The potential of antibiofilm drugs has been explored in different research fields seeking the development of a promise in the dispersion of these drugs. In this way, the investigation of possibilities beyond common antibiotics has given space to natural compounds, exploring new prospects with a lower resistance of these bacteria to drugs [44]. The growing interest in the pharmacology, biochemistry and chemistry of quinone-based antibiotics in the fight against the biofilm generated by bacteria is not new precisely because of the intrinsic characteristics of quinone compounds [23].

Anthraquinones are widely found in nature, having great medicinal importance as potential drug candidates as antimicrobial, antitumor, antiviral and also anti-inflammatory agents [45]. Bacterial infection is a major concern in medicine due to the increase in mutations and biofilm-forming microorganisms. Almost 80% of bacterial infections are attributed to biofilms and represent an increase in medical costs as the biofilm provides a reservoir for bacteria and acts as a source of chronic infection in the human body. Streptococcus mutans is one of the rapidly multiplying bacteria that is present in the mouth and is responsible for initiating dental infections related to biofilms. *Staphylococcus aureus* is another common pathogen commonly found in wounds and other infections, and it also causes biofilms [45]. It is also associated with infections related to permanent medical devices such as catheters [46]. Thus, any chemical agent that can inhibit the formation or eradicate biofilms would be useful to treat different infections.

Farooq et al. [46] evaluated three new alkyls substituted anthraquinone derivatives, commonly called symploquinones, which were isolated from *Symplocos racemosa.* The structures of symploquinone A-C (compounds **52**–**54**) are shown in Figure 10.

These compounds were then subjected to antibacterial or antibiofilm tests. The results showed that the isolation of chemical constituents from *Symplocos racemosa* resulted in the successful isolation of three alkyl substituted anthraquinones whose structures were elucidated using different spectroscopic techniques. These compounds were found to significantly inhibit *Staphylococcus aureus* and *Streptococcus mutans* biofilm formation at minimal inhibitory concentration (MIC) or sub-MIC concentrations. With that, these compounds showed a potential to be used as an alternative to different drugs and/or in combination with different compounds to reduce drug resistance.

According to Campanini-Salinas et al. [47], compounds that have the structure of the quinone ring, known for their antimicrobial, antibacterial and other properties, still need to be explored regarding the knowledge of the full spectrum of their actions and their derivatives. The authors synthesized 17 thiophenyl quinone derivatives. The derivatives were entirely eggs, differing from each other by the added chemical groups and by the position of the carbon from the bond with the thiophenyl ring. The compounds developed showed inhibition of the growth of multidrug-resistant *Staphylococcus aureus* and *Enterococcus faecium*, with MIC values between 1 and 32 μg/mL. No toxicity to mammalian cells was identified. A comparison of the bactericidal kinetics of the quinone compounds with a commercial antibiotic (vancomycin) was made. Results demonstrated that quinone derivatives reduced 99.9% of the different bacterial populations and were studied two times faster than vancomycin. The study showed a comparative advantage in the result, even more so when one observes the contributions of the literature on the antibacterial capabilities of quinones.

The effects of natural compounds on biofilm formation have been extensively studied, in order to identify antagonists of biofilm formation at sublethal concentrations [48]. Salicylic and cinnamic acids are some examples of these compounds that interact with the quinone oxidoreductase WrbA, a potential modulator of biofilms and a biomarker of the antibiofilm compound. However, the role of WrbA in biofilm development is still poorly understood [26]. Rossi et al. [49] investigated the role of the WrbA compound in biofilm maturation and its oxidative stress, using DwrbA wild-type and mutant *Escherichia coli* strains. As a result, the functional effectiveness of WrbA as a molecular target of salicylic and cinnamic acids was verified. The lack of WrbA did not impair planktonic growth, but directly affected biofilm formation through a mechanism that depends on ROS. Endogenous oxidative events in the mutant strain generated a stressful condition to which the bacterium responded by increasing catalase activity to compensate for the lack of WrbA. Cinnamic and salicylic acids inhibited the quinone oxidoreductase activity of purified recombinant WrbA. Thus, the effects of these antibiofilm molecules on WrbA function were proven for the first time. Figure 1 shows the inhibition effects of cinnamic and salicylic acids on the function of WrbA.

In a recent study, J. H. Lee et al. [50] also studied the potential synergistic effect of specific adjuncts [51] in combination with anthraquinone-based antibiotics on biofilm inhibition. The authors reported efficient biofilm growth inhibition (by ≥70% versus non-treated controls) on the *Staphylococcus aureus* strain (MSSA 6538) for alizarin-2 at a concentration of 10 µg/mL, as well as for purpurin-14, emodin-15 and quinalizarin-16 (Figure 3), whereas for seven other anthraquinone-like derivatives, no significant effect was observed at the same concentration (Figure 2).

Interestingly, further experiments conducted on bacteria cultures of both *Staphylococcus aureus* strains, MSSA 25923 and MRSA MW2, demonstrated similar behavior with alizarin-2, whereas a concentration of 50 µg/mL is required to inhibit biofilm growth by ≥70% of *Staphylococcus epidermidis* (ATCC 14990). Moreover, alizarin-2 remains ineffective to Gram-negative bacteria strains (*Escherichia coli* O157:H7 and *Pseudomonas aeruginosa* PAO1) even at concentrations of up to 100 µg/mL. They ascribed these remarkable antibiofilm properties against *Staphylococcus aureus* to the presence of hydroxyl groups at the C-1 and C-2 position of the anthraquinone skeleton and their inherent ion chelation properties (Figure 11).

Indeed, recent studies have evidenced the important role played by Ca^2+^ in biofilm growth inhibition on the *Staphylococcus aureus* strain [52]. This alizarin-adjuvant was envisioned to enhance this effect through synergistic action. Zhang and Ma [53] point out that the combination of cationic quinone analogs with naphthoquinone or anthraquinone allows the nuclei to present two different modes of action, with excellent antibacterial activity against *Staphylococcus aureus* (MIC ¼ 0.032 and 0.064 mg/mL). Its action helps to inhibit the redox effects on bacteria, which may be the cause of antibacterial effects. The antibiofilm activity observed in quinones is due to six main mechanisms: substrate deprivation, membrane rupture and binding to the adhesin complex to the cell wall, protein binding, interaction with eukaryotic DNA and blockage of viral fusion [54].

Silva et al. [55] analyzed the antibiofilm action of benzoquinone oncocalyxone A. The study used *Staphylococcus epidermidis* 70 D and *Staphylococcus aureus* MED 55 considering their importance in biofilm formation. Activity analysis was based on % of control values, indicating that 9.43 μg/mL onco-A significantly inhibited biofilm production by ~70% in the methicillin-resistant *Staphylococcus aureus* MED 55 strain. Regarding hemolytic activity, onco-A did not present hemolytic activity at the concentrations tested in the inhibition of bacterial growth.

Yang et al. [56] demonstrated that the use of emodin (anthraquinone), obtained from *Rheum palmatum L*. through in vitro study, was effective against pathogens of *Candida albicans*, *Candida krusei*, *Candida parapsilosis*, *Candida tropicalis* and *Staphylococcus aureus*. There was inhibition of biofilm formation through cell targeting of kinase signaling, through action in the planktonic cell from the reduction of hyphal formation. The inhibitory concentration showed values of MIC = 12.5 µg/mL, MFC = 25 µg/mL and MIC/MFC = 25 µg/mL.

The use of natural antibiofilm agents, which selectively kill persistent biofilms, allows the diffusion of antimicrobial agents from these compounds in the biofilm matrix. The main target of these natural compounds is the phases of the biofilm cycle, intending to degrade the matrix and, finally, destroy the released cells [57]. Although they show promising results regarding antibacterial action and inhibition of biofilms, these compounds are still lacking in clinical studies. This is due to their low performance, cytotoxic and hemolytic activities and the side effects generated in individuals, such as kidney injuries and damage to the nervous system [58].

Recent studies have contributed to providing additional information on the mechanism of action of quinones. [59]. However, it is very likely that this mechanism of action is not general but must be structure-like and dependent on similar pathways; likewise, the potential for membrane potential dissipation and their acting as respiratory chain inhibitors need to be further investigated.

## 4. Quinone-Based Hydrogels: New Applications

Tissue engineering materials typically require the employment of three-dimensional (3D) fiber structure supports to exhibit excellent properties or other biological functions [60]. For example, many soft tissues, such as skin and ligaments, have excellent load-bearing performance due to their fiber support structures, even with a “soft and wet” nature. Hydrogels, networks of hydrated three-dimensional polymers, have attracted great attention due to their “soft and wet” nature that is similar to biological tissues. Nature presents many examples of biogenic substances that provide inspiration for the development of advanced structural materials [61].

Quinones are reactive compounds that undergo non-enzymatic reactions with various nucleophiles. The uses of quinones in nature include the hardening process of insect cuticle sclerotization or tanning. Specifically, the hardening is a result of the covalent crosslinking of amino groups present on the protein or chitin chains by quinone rings, which are derived from the oxidation of phenol. This oxidation creates crosslinks that give the cuticle enhanced mechanical properties [62]. Chen et al. [63] evaluated an alternative for the preparation of cross-linked quinones hydrogels based on chitin nanofibers (ChNF) using amino groups, inspired by the quinone hardening process during sclerotization of the insect cuticle. By increasing the number of amino groups on the crystalline surface of chitin through deacetylation, the resulting surface deacetylated chitin nanofiber (S-ChNF) turns into a dark hydrogel when reacted in hydroquinone (HQ)/copper (Cu(II)) solutions. The results of the research show that the S-ChNF-based hydrogel showed nearly 10 times greater tensile strength than the ChNF-based hydrogel due to extended cross-linking effect between the quinone and amino groups. Given the natural sustainability of chitin and the demonstrated improved mechanical strength, the study showed a promising strategy to manufacture a bioinspired based on S-ChNF hydrogel for potential applications in biomedical engineering fields.

Because of the great availability of natural sources with quinone compounds, an increasing range of studies has sought to expand their applicability in the field of bioengineering. Its intrinsic properties are appealing, as well as their structural diversity, therapeutic potential and the mechanisms of action that we could observe in these compounds [17]. The combination of mechanical properties that can be observed in these compounds include stiffness, toughness and hardness [64]. This attention is associated with the use of quinone compounds in the production of hydrogels, mainly aiming to reduce their limitations regarding low solubility in common solvents, poor mechanical properties and difficulties related to their manufacture.

Many adhesive hydrogels can be developed based on different adhesion strategies, such as supramolecular-based adhesives, hydrogen bonds, nucleobases and topological adhesive hydrogels. Adhesive hydrogels are typically prepared, most often, through free radical polymerization, requiring external stimuli such as UV irradiation or thermal initiation to induce gelation, which limits their practical applications. Therefore, it is extremely important to develop new routes for the production of adhesive hydrogel bioelectronics, especially if they are implantable bioelectrodes, which can undergo self-fixation triggering free radical polymerization without external stimuli under physiological conditions [65].

Natural adhesion has also inspired the development of adhesive hydrogels. For example, inspired by the mussel adhesive, catechol-based molecules, peptides and polymers have been used to modify the surfaces of hydrogel materials [65,66]. Studies have shown that adhesive hydrogels can be produced by a mussel-inspired strategy [66,67]. However, based on studies inspired by the adhesion of mussels, it is critical to control the redox balance between the catechol and quinone groups, where a high content of catechol groups is maintained within the hydrogel networks to provide greater adhesion to the material [68,69].

In the studies by Gan et al. [69], the use of silver-lignin nanoparticles (Ag-lignin NPs) triggered the catechol redox chemical reaction. Ag-lignin NPs generate free radicals that are used to promote free radical polymerization of monomeric acrylates such as acrylic acid and poly (ethylene glycol) diacrylate to form a hydrogel capable of tissue adhesion, originating from lignin within the redox chemistry of catechol group. A schematic of the chemical reaction of marine mussel catechol with the combined oxidative decarboxylation and quinone-catechol redox catalysis reactions used to form the lignin-based adhesive antimicrobial hydrogel is shown in Figure 3.

The research developed by Chen et al. [63] points to quinone derivatives as effective crosslinkers in the manufacture of high-strength hydrogels. The quinone-crosslinked hydrogel developed by Chen et al. [70] demonstrated significantly improved tensile performance. From the control of the reaction time, the group found out that the tensile strength can reach up to 2.96 MPa, presenting biocompatible hydrogel with sustainable properties. Such discoveries allow the use of quinone-crosslinked hydrogel in biomedicine and agriculture.

Khan et al. [71] used lignin as a redox polymer, due to its diverse phenolic compounds that, when electrochemically oxidized, can form quinones. Quinones can also be efficient as a means of transporting electrons in organisms through reversible redox processes. In acidic aqueous solutions, quinone/hydroquinone pairs can provide a reversible 2-electron, 2-proton (2e^−^/2H^+^) redox reaction process in a single step (Figure 12). Quinones are easily reduced to hydroquinones, and can even be oxidized to quinones, generating the redox system [6].

The characteristics that make up the redox reversible processes of quinones also attracts attention to the generation of hydrogels as high-performance pseudocapacitive electrodes. High pseudocapacitance and strong adhesion to carbon nanostructures through non-covalent interactions are characteristics that improve the attractiveness of the material. However, studies in this direction need to explore the low conductivity and short life cycle of quinone-containing hydrogels [72].

Jia et al. [73] investigated the use of phenol-quinone in the study of adhesive hydrogels. They verified that acid chelated-Ag (TA-Ag) nanozyme with peroxidase (POD)-like activity presented a dynamic redox balance derived from phenol-quinone. It provides the hydrogels with repeatable adhesiveness, similar to the accession of mussels. In addition, the ultrasmall hydrogel also showed antibacterial activity, allowing its use in accelerating tissue regeneration, implantation and for preventing infections. The literature points out the fundamental role of quinone and its derivatives as crosslinkers in the reactions, indicating that its properties are essential for hydrogels’ development with improved properties. The improvement of hydrogels can provide more comfortable medical experiences and reduce the number of interventions needed in procedures. Understanding the dynamics of effects observed by the use of quinones, there are chemical, typological and mechanical effects that are quite beneficial in their application in hydrogels [74].

Zhang et al. [53] highlight that quinone can be used in the crosslinking of chitosan to develop a wet film that has high strength. The use of quinone derivatives is promising, especially because of their reactivity with other products, such as citric acid (CA), acrylic acid (AA) and poly (acrylamide-co-acrylic acid (P(AAm-co-AA)), which have all shown results in the gelation of hydrogel solutions with promising properties. It was possible to observe adhesiveness to different substrates, such as human skin, wood, glass, metal, plastic and ceramic, with varying adhesion strength: 6.13 kPa on glass, 3.59 kPa on plastic surfaces and 5.10 kPa on metals. Antimicrobial activity and ultra-stretchability were also observed, as per research by He et al. [75].

## 5. Conclusions

The literature points to strong evidence regarding the antibacterial/antifungal activity of quinones. It may cause microbes to develop increased resistance to biocides in addition to the cumulative side-effects of drugs and may have a substantial impact on the patient’s microbiota ecology. Literature survey shows that quinone compounds have high efficacy against Gram-positive and Gram-negative bacteria, as well as good selectivity for cells. The main appeal of quinones’ use in the development of new antibiotics and hydrogels lies in their antibiofilm action, since in vitro studies show good minimum inhibitory concentration results. In addition, the rapid bactericidal activity and low possibility of inducing drug resistance provides an attractive approach for the use of quinones to resolve the phenomenon of bacterial multidrug resistance. They require further studies regarding their mechanisms of action, metabolism and cytotoxicity in normal cells.

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
