# Peer review of "Quinones as an Efficient Molecular Scaffold in the Antibacterial/Antifungal or Antitumoral Arsenal"

_ijms, 2022, doi:10.3390/ijms232214108_

Round 1
Reviewer 1 Report
The authors have provided a review of Quinone-based compounds and their wide applications against Gram-positive and Gram-negative microbial infections. The article could be accepted after addressing the following points:
As the authors mentioned that the Quinone-based compounds have shown long date unwavering efficiency against both Gram-positive and Gram-negative microbial infections, but they could not provide the applications from the literature. The authors should do the better literature review to cover the subject
2) Characterization part of the Quinone-based compounds is missing. The authors should provide some morphological and structural data in order to provide the physical and chemical properties of different Quinone-based materials and their dimensions in used form.
3) The reference citing has some problems. It shows the reference error in the saved pdf file. The authors should address this issue.
4) Authors should provide the literature about the application data where one can understand the working mechanism of Quinone-based materials against bacteria.
Author Response
Reviewer 1
The authors have provided a review of Quinone-based compounds and their wide applications against Gram-positive and Gram-negative microbial infections. The article could be accepted after addressing the following points: As the authors mentioned that the Quinone-based compounds have shown long date unwavering efficiency against both Gram-positive and Gram-negative microbial infections, but they could not provide the applications from the literature. The authors should do the better literature review to cover the subject
Reply : The article discusses the latest achievements in the last five years. For a broader view of the field readers may refer to other journals, for example, ref. 8 and the references cited therein.
---------------------------------------------------------------------------------
2) Characterization part of the Quinone-based compounds is missing. The authors should provide some morphological and structural data in order to provide the physical and chemical properties of different Quinone-based materials and their dimensions in used form.
Reply : Additional information have been provided in the body text to give structure-properties relationships along additional examples.
---------------------------------------------------------------------------------
3) The reference citing has some problems. It shows the reference error in the saved pdf file. The authors should address this issue.
Reply : The references have been checked and listed appropriately to generate an error-free pdf.
---------------------------------------------------------------------------------
4) Authors should provide the literature about the application data where one can understand the working mechanism of Quinone-based materials against bacteria.
Reply : To support the possible mechanism of quinones derivatives against bacteria a reference has been added with a sentence in the body text. “Chen, B.-C.; Ding, Z.-C.;Dai, J.-S.; Chen, N.-P.; Gong, X.-W.; Ma, L.-F.; Qian, C.-D. New insights into the antibacterial mechanism of cryptotanshinone, a representative diterpenoid quinone from Salvia miltiorrhiza Bunge. Front. Microbiol. 2021, 12:647289."
Reviewer 2 Report
Some compounds with quinones scaffold esp naphthoquinones also showed potent anti-fungi activity, but it cannot be included only into antibacterial arsenal, so please think again about the title, and hope the title can cover anti-fungi activity.
Actually from line 147, many sentence about antifungi.
Part 1. About Introduction
The paper is discussing about Quinones: an efficient molecular scaffold in the antibacterial arsenal, so, please delete sentences which are no relation with the theme, for example in Line 31 and 59. Etc.,
Line 31, “Quinone derivatives have prominent applications in several areas, such as chemosensor, catalyst, dyes, energy storage, electron transfer, among others [6,7]”
Line 59 “Moreover, their inherent properties make them systems of choice for industrial dye-stuffs, energy storage purposes in batteries [11]”
Please cut the branch in the theme tree and revise the introduction part as succinct as possible then reader can focus on the theme on antibacterial properties,
Line 66, Streptomyces caeruleorubidus----Streptomyces coeruleorubidus and italic please
Part 3. Quinone-Base Antibiotics with Anti-Biofilm Action
1, Please delete all unnecessary sentence such as Line 192-line 197 including in addition.
Start from “Bacteria and fungi have developed strategies, including the formation of biofilms to prevent the action of many inhibitors……..OR please delete the whole first paragraph.
In the Part of 3, please also use academic words instead of colloquial language.
Author Response
Reviewer 2
Some compounds with quinones scaffold esp naphthoquinones also showed potent anti-fungi activity, but it cannot be included only into antibacterial arsenal, so please think again about the title, and hope the title can cover anti-fungi activity. Actually, from line 147, many sentence about antifungi.
Reply : As pointed out by the reviewer the title has been changed accordingly to cover the different quinones activities discussed in the paper.
---------------------------------------------------------------------------------
The paper is discussing about Quinones: an efficient molecular scaffold in the antibacterial arsenal, so, please delete sentences which are no relation with the theme, for example in Line 31 and 59. Etc., Line 31, “Quinone derivatives have prominent applications in several areas, such as chemosensor, catalyst, dyes, energy storage, electron transfer, among others [6,7]” Line 59 “Moreover, their inherent properties make them systems of choice for industrial dye-stuffs, energy storage purposes in batteries [11]” Please cut the branch in the theme tree and revise the introduction part as succinct as possible then reader can focus on the theme on antibacterial properties,
Reply: These sentences have been retained to demonstrate the broad application of quinone derivatives in which the electron transfer process is important and possibly involved in the antibacterial, antifungal or antitumoral mechanism of action of these compounds.
---------------------------------------------------------------------------------
Line 66, Streptomyces caeruleorubidus----Streptomyces coeruleorubidus and italic please
Reply: the name of the above bacteria has been corrected and written in italic in the body text.
---------------------------------------------------------------------------------
Part 3. Quinone-Base Antibiotics with AntiBiofilm Action
1, Please delete all unnecessary sentence such as Line 192-line 197 including in addition.
Start from “Bacteria and fungi have developed strategies, including the formation of biofilms to prevent the action of many inhibitors……..OR please delete the whole first paragraph. In the Part of 3, please also use academic words instead of colloquial language.
Reply : Thanks to the reviewer we have deleted all unnecessary sentences and almost all the first paragraph in this part 3.
Reviewer 3 Report
The authors Marcos Aurelio Dahlem Junior, Ronald W. Nguema Edzang, André Luis Catto, and Jean-Manuel Raimundo submitted the manuscript entitled “Quinones: an efficient molecular scaffold in the antibacterial arsenal” to the journal “International Journal of Molecular Sciences” in order to be considered for publication as a “Review”.
The manuscript provides an overview of exemplary compounds with quinones partial structure that have antibacterial properties. However, contrary to the title, antifungal and antitumoral aspects are also discussed. Below are some comments on the manuscript.
Referring to Scheme 1, there are several mistakes and concerns to mention: (4) mitomycin C, please change the way of how the “amino” groups are connected, i.e. the nitrogen atom and not the hydrogens (also (2) alizarin: one of the hydroxyl groups); (5) verqoliquinone A ?; (8) geldanamycin, please check for E/Z-configuration; (6) ubiquinones, it is two times methoxy groups and not methyl groups; (general) please be consistent how to present methyl/methoxy groups, i.e. Me, CH3, or without further declaration. The authors are kindly asked to revise all the chemical structures also given in the other Schemes/Figures.
“For instance, they play important roles in many enzymatic and physiological systems due to their central role as redox mediator in several electron-transfer processes (Scheme 1)” I guess that does not apply for all of the compounds given in Scheme 1. However, it does not become clear which of the compounds fulfill which particular effects. Therefore, it is suggested to develop on that point.
“…leading to multidrug resistances” Personally, I would list a few more aspects here as examples that lead to the development of resistance, such as: efflux pumps (e.g., https://doi.org/10.1186/s12866-021-02250-x), porin protein loss and decreased drug uptake (e.g., https://doi.org/10.1128/spectrum.00148-22) among others (e.g., https://doi.org/10.1016/j.micpath.2021.104915)
“Due to their unique pharmaceutical applications, quinones are widely used as anticancer, antioxidant, antimalarial, antimicrobial, anti-inflammatory agents”. Maybe it is more suitable to cite a review dealing with quinones in medicinal chemistry, such as 10.2174/1573406416666201106104756.
“o-benzoquinone and p-benzoquinone” Maybe it is better to term them 1,2- and 1,4-benzoquinone.
In the introduction, some aspects are repeated, such as the diverse use of quinones in pharmaceutical / industrial application. This part could be compressed.
“It is verified that active sites in quinones and derived compounds allow the development of molecules with better performance in the fight against diseases.” What does that mean in particular? Can the authors please provide examples, literature to somehow confirm this statement.
“Streptomyces caeruleorubidus” in italics, also “in vitro” and “in vivo”, “et al.” Also other formal inconsistencies such as punctuation, superscript, use of dash or not: anti-… vs. anti…
Scheme 2: The chemical structure of doxorubicin is wrong (additionally the “hydrochloride” is missing). Please revise.
Please revise the order of numbering in the text and adjust in the figure: NOT: …17, 18, 21, 19, 20. BUT: …17, 18, 19, 20, 21.
“…using the discovered properties of quinones, doxorubicin and nitroazoles to provide practical solutions in oncological radiotherapy treatments and pathophysiology” Is such information really relevant? The subject of the review is "quinones" and their "antibiotic arsenal".
“Furthermore, the widespread use of antibiotics as well as their overusing also led the increase of the bacterial resistance striven the scientific community to search urgently new and more effective derivatives and strategies”. This was already briefly noted at the beginning of the introduction. Basically, the introduction lacks a clear structure because it jumps back and forth between different aspects.
“Other antibiofilm approaches based on the inhibition of bacterial adhesion or proliferation have also prompted strong interest [23]. Those include…”The question is whether quinones are really used here. The introduction seems to me to be poorly focused in some places. Also: “Several natural naphthoquinones for instance … have biological and pharmacological properties”. Is it also antibacterial properties they exhibit? Also: Futuro et al. … antifungal is not equal with antibacterial.
At the end of the introduction, a concrete question/topic, the goal of the review would be desirable.
manly – mainly
“that can contribute to enlarge the scope and biocidal activities” What does that mean in particular? How does this take place?
“due to the intrinsic characteristics of quinone compounds” What does this mean in particular? Can the authors provide some examples?
In my opinion, it is not balanced in this manuscript to report so extensively on the nature of biofilms.
It is not clear from Scheme 9 which of the structures represents symploquinone A, B, C.
The abbreviation “ROS” is introduced twice.
Several times in the manuscript: [Error! Reference source not found.] Please revise.
“…of the active anthraquinones 2, 10-12” However, the compounds are labelled with 48, 49, 50. That is confusing, strange. Please revise.
Conclusion: “The literature points to strong evidence regarding the antifungal activity of naphthoquinones.” The review should actually deal with antibacterial effects.
“We also verified their wide use in the most varied activities” What did the verification look like? Only examples from the literature were taken up. I think it is not correct to speak of verification here.
Basically, I miss the common thread in the manuscript. There is a lot of jumping back and forth between different contents. The focus on the antibacterial effects is not given. Again and again, antifungal properties are brought into play. To me, the manuscript does not seem balanced. In some places it goes into great detail, elsewhere it is superficial. Often the data are not critically put into context, but merely listed narratively. In my opinion, it is not clearly worked out why quinone derivatives in particular are promising for the design of antibacterial drugs. The issue of differentiating between trating Gram-positive and Gram-negative bacteria is only touched upon. It is here that drug design faces a challenge. Whether this can be achieved by quinones is not answered in the review. Formally, the manuscript also has much room for improvement. In addition, there are frequent repetitions of content, which can be annoying to the reader.
All of these points ultimately lead me to not endorse the further processing of the manuscript in a high impact journal (IF=6.2). I hope my comments show the authors where there is still significant potential for improvement of their manuscript. All the best!
Author Response
Reviewer 3
The authors Marcos Aurelio Dahlem Junior, Ronald W. Nguema Edzang, André Luis Catto, and Jean-Manuel Raimundo submitted the manuscript entitled “Quinones: an efficient molecular scaffold in the antibacterial arsenal” to the journal “International Journal of Molecular Sciences” in order to be considered for publication as a “Review”. The manuscript provides an overview of exemplary compounds with quinones partial structure that have antibacterial properties. However, contrary to the title, antifungal and antitumoral aspects are also discussed. Below are some comments on the manuscript.
Reply : Considering the remark of the reviewer the title has been changed accordingly to cover the different quinones activities discussed in the paper.
---------------------------------------------------------------------------------
Referring to Scheme 1, there are several mistakes and concerns to mention: (4) mitomycin C, please change the way of how the “amino” groups are connected, i.e. the nitrogen atom and not the hydrogens (also (2) alizarin: one of the hydroxyl groups); (5) verqoliquinone A ?; (8) geldanamycin, please check for E/Z-configuration; (6) ubiquinones, it is two times methoxy groups and not methyl groups; (general) please be consistent how to present methyl/methoxy groups, i.e. Me, CH3, or without further declaration. The authors are kindly asked to revise all the chemical structures also given in the other Schemes/Figures.
Reply : The structures have checked and corrected in whole document.
---------------------------------------------------------------------------------
“For instance, they play important roles in many enzymatic and physiological systems due to their central role as redox mediator in several electron-transfer processes (Scheme 1)” I guess that does not apply for all of the compounds given in Scheme 1. However, it does not become clear which of the compounds fulfill which particular effects. Therefore, it is suggested to develop on that point.
Reply: According to the reference cited, this condition applies to plastoquinone and ubiquinone in living organisms. The text has been modified to show this particularity.
---------------------------------------------------------------------------------
“…leading to multidrug resistances” Personally, I would list a few more aspects here as examples that lead to the development of resistance, such as: efflux pumps (e.g., https://doi.org/10.1186/s12866-021-02250-x), porin protein loss and decreased drug uptake (e.g., https://doi.org/10.1128/spectrum.00148-22) among others (e.g., https://doi.org/10.1016/j.micpath.2021.104915)
Reply: We perfectly agree, and additional references have been added in the first paragraph of the introduction in order to consider the additional aspects recommended by the reviewer.
---------------------------------------------------------------------------------
“Due to their unique pharmaceutical applications, quinones are widely used as anticancer, antioxidant, antimalarial, antimicrobial, anti-inflammatory agents”. Maybe it is more suitable to cite a review dealing with quinones in medicinal chemistry, such as 10.2174/1573406416666201106104756.
Reply : The reference suggested by the reviewer, which deals with quinones applied to the field of medicinal chemistry, was added to the text of the manuscript.
---------------------------------------------------------------------------------
“o-benzoquinone and p-benzoquinone” Maybe it is better to term them 1,2- and 1,4-benzoquinone.
Reply: The ortho- and para-benzoquinone have been termed as 1,2- and 1,4-benzoquinone respectively.
---------------------------------------------------------------------------------
In the introduction, some aspects are repeated, such as the diverse use of quinones in pharmaceutical / industrial application. This part could be compressed.
“It is verified that active sites in quinones and derived compounds allow the development of molecules with better performance in the fight against diseases.” What does that mean in particular? Can the authors please provide examples, literature to somehow confirm this statement.
Reply: The section has been revised. The quoted text was removed, considering that it can be seen in other sections of the article.
---------------------------------------------------------------------------------
“Streptomyces caeruleorubidus” in italics, also “in vitro” and “in vivo”, “et al.” Also other formal inconsistencies such as punctuation, superscript, use of dash or not: anti-… vs. anti…
Reply : All words needed to be in italic have been reviewed and corrected accordingly as well as punctuation etc.
---------------------------------------------------------------------------------
Scheme 2: The chemical structure of doxorubicin is wrong (additionally the “hydrochloride” is missing). Please revise.
Please revise the order of numbering in the text and adjust in the figure: NOT: …17, 18, 21, 19, 20. BUT: …17, 18, 19, 20, 21.
Reply : the structure of doxorubicin has been corrected as requested; the order of numbering of the schemes and figures as well.
---------------------------------------------------------------------------------
“…using the discovered properties of quinones, doxorubicin and nitroazoles to provide practical solutions in oncological radiotherapy treatments and pathophysiology” Is such information really relevant? The subject of the review is "quinones" and their "antibiotic arsenal".
Reply : As the title has been modified to cover other activities of quinones we keep this part and additional references to the use of quinones in the area of antibiotics were added.
Huigens, R.W, Yang, H, Liu, K, Kim, Y.S, Jin, S. An ether-linked halogenated phenazine-quinone prodrug model for antibacterial applications. Org. Biomol. Chem., 2021, 19, 6603-6608.
Hegedűs, D, Szemerédi, N, Spengler, G, Szatmári, I. Application of partially aromatic ortho-quionone-methides for the synthesis of novel naphthoxazines with improved antibacterial activity. Eur. J. Med Chem, 2022, 237, 114391
---------------------------------------------------------------------------------
“Furthermore, the widespread use of antibiotics as well as their overusing also led the increase of the bacterial resistance striven the scientific community to search urgently new and more effective derivatives and strategies”. This was already briefly noted at the beginning of the introduction. Basically, the introduction lacks a clear structure because it jumps back and forth between different aspects.
Reply: The introduction has been revised. However, certain elements were evidenced due their weight in the analysis. It is expected that, in this new format, the introductory section will be more assertive.
---------------------------------------------------------------------------------
“Other antibiofilm approaches based on the inhibition of bacterial adhesion or proliferation have also prompted strong interest [23]. Those include…”The question is whether quinones are really used here. The introduction seems to me to be poorly focused in some places. Also: “Several natural naphthoquinones for instance … have biological and pharmacological properties”. Is it also antibacterial properties they exhibit? Also: Futuro et al. … antifungal is not equal with antibacterial.
Reply: Yes, quinones are included and some of these studies cited that address antibiofilms and the inhibition of bacterial proliferation in some areas of application. The title has been changed to cover in certain case the antifungal properties of the quinones derivatives.
---------------------------------------------------------------------------------
At the end of the introduction, a concrete question/topic, the goal of the review would be desirable.
Reply: As requested, a central question that guides the article was incorporated. The question aims to establish the relationship between all the elements highlighted and aims to expose the intention of the paper to establish a literature review on the subject. “With these considerations in mind, this article aims to identify relevant properties of quinones for their use as molecular scaffolds.”
---------------------------------------------------------------------------------
manly – mainly
Reply : The correction of this word was made accordingly.
---------------------------------------------------------------------------------
“that can contribute to enlarge the scope and biocidal activities” What does that mean in particular? How does this take place?
Reply : According to the reference cited, there is an influence on the structure of the protein, which interferes with the regulatory capacity for some conditions. This explanation has been incorporated into the text to ensure understanding of the sentence.
---------------------------------------------------------------------------------
“due to the intrinsic characteristics of quinone compounds” What does this mean in particular? Can the authors provide some examples?
Reply: The intrinsic characteristics of quinones refer to their nature, which lead to chemical properties of particular importance, such as being largely oxidizing and electrophilic. These properties, for example, allow them to participate in redox cycle reactions and nucleophilic Michael addition reactions, respectively. These examples were added to the manuscript text.
---------------------------------------------------------------------------------
Conclusion: “The literature points to strong evidence regarding the antifungal activity of naphthoquinones.” The review should actually deal with antibacterial effects.
Reply: We agree with the reviewer hence the manuscript has been revised and more discussions and studies have been added regarding the antibacterial effects of quinones but also includes antifungal examples. The sentence has been changed accordingly.
---------------------------------------------------------------------------------
It is not clear from Scheme 9 which of the structures represents symploquinone A, B, C.
Reply : In order to clarify the understanding of Scheme 9, the respective letters (A, B and C) were added to the structures.
---------------------------------------------------------------------------------
The abbreviation “ROS” is introduced twice.
Reply : the second explanation in the text of ROS was removed.
---------------------------------------------------------------------------------
Several times in the manuscript: [Error! Reference source not found.] Please revise.
Reply: This error was checked and corrected throughout the manuscript.
---------------------------------------------------------------------------------
“…of the active anthraquinones 2, 10-12” However, the compounds are labelled with 48, 49, 50. That is confusing, strange. Please revise.
Reply: Appropriate correspondence between the text and the labelled compounds 48, 49, 50 was made.
---------------------------------------------------------------------------------
“We also verified their wide use in the most varied activities” What did the verification look like? Only examples from the literature were taken up. I think it is not correct to speak of verification here.
Reply: the conclusion has been revised and rewritten accordingly.
Round 2
Reviewer 1 Report
Recommendation to accept in present form
Author Response
We thanks for referee for his final decision
Reviewer 2 Report
The authors already revised according to suggestion, but The special issue is Antibacterial Strategies in Biomaterials: Current Progress and Challenges. Now the whole manuscript changed name as "Quinones as an efficient molecular scaffold in the antibacterial/antifungal or antitumoral arsenal", which is far from antibacterial Strategies. Actually, antifungal and antitumor is unacceptable, authors should focused on the theme in the very beginning when collect papers to reflect the progress.
Author Response
First of all we thanks the review for his additional comment. Reply : The main focus of the manuscript "Quinones as an efficient molecular scaffold in the antibacterial/antifungal or antitumor arsenal" remains to be in the area of antibacterial strategies, although the title has been slightly changed. The revision and change of the title was (i) suggested by one of the reviewer (ii) and also because several antibacterial quinones derivatives exhibit potent efficacies as antifungal or antitumoral compounds. Nevertheless, although antifungal and antitumoral properties are addressed herein the heart of the text and manuscript concerns the antibacterial properties of quinones. Besides, one of the reviewers has suggested, from the first version of the manuscript, to cover others drug aspects of the quinones derivatives thus the text and title were modified accordingly to fulfill and cover the requirements Even with a broader title the manuscript fits into the special issue “Antibacterial Strategies in Biomaterials”, as the main focus remains as antibacterial, with additional applications such as antifungal or antitumoral. .Reviewer 3 Report
The authors provided a revised version of the manuscript. They acted on every concern and suggestion as mentioned before.
The title of the article was slightly modified, now also considering antifungal and antitumoral activities of quinones. This is, of course, an easy solution to get around the whole problem with the "antifungal" parts of the manuscript. However, I see the following concerns: on the one hand, the manuscript is not comprehensive enough to include all (at least a representative amount) antifungal and anticancer examples of quinones. On the other hand, the special issue is on “Antibacterial Strategies in Biomaterials”. The manuscript is a limited fit with the special issue with this new title. But this should rather be decided by the guest editors.
The authors corrected the wrong chemical formula and adapted chemical nomenclature. This seems fine now. Elsewhere, they have made minor changes to some of their statements so that now the statements appear correct in context. Or they added the information necessary to get the meaning. Formal inconsistencies were also optimized. A few formal trifles (e.g. reference [0], spaces, upper and lower case) would then have to be improved in the proof.
They also incorporated recent literature on the one hand on quinones in medicinal chemistry and on the other hand brief insights on mechanisms of bacterial resistance development. This makes the introduction somewhat smoother in my opinion. The authors keep their introduction in this form as far as possible. It's not really bad, but in some places I would expect it to be a bit more condensed. But that is a personal opinion. On the other hand, the authors argue that in this form they make the problems of antibiotic resistance and the potential of quinones even more clear. Of course, you can look at it that way.
Author Response
We thanks the reviewer for his positive agreement and also for his comments and remarks concerning the minor errors. The manuscript was revised accordingly and corrected where necessary.
Round 3
Reviewer 2 Report
The authors already revised according to suggestion.
Author Response
We thanks the reviewer for all of his fruitful comments.
Reviewer 3 Report
The manuscript “Quinones as an efficient molecular scaffold in the antibacterial/antifungal or antitumoral arsenal” is generally suitable to be published in the journal “Int. J. Mol. Sci.” as it matches the scope “fundamental theoretical problems of broad interest in biology, chemistry and medicine”. There are some few minor concerns as mentioned before. Once the authors acted on them, the manuscript can be considered for potential publication.
Author Response
We thanks the reviewer for his comments. The manuscript has been revised accordingly to his requirements.